# Diagnosis of Human Leptospirosis: Comparison of Microscopic Agglutination Test with Recombinant LigA/B Antigen-Based In-House IgM Dot ELISA Dipstick Test and Latex Agglutination Test Using Bayesian Latent Class Model and MAT as Gold Standard

**DOI:** 10.3390/diagnostics12061455

**Published:** 2022-06-13

**Authors:** Sujit Kumar Behera, Thankappan Sabarinath, Balasubramanian Ganesh, Prasanta Kumar K. Mishra, Roshan Niloofa, Kuppusamy Senthilkumar, Med Ram Verma, Abhishek Hota, Shanmugam Chandrasekar, Yosef Deneke, Ashok Kumar, Muruganandam Nagarajan, Deepanker Das, Sasmita Khatua, Radhakrishna Sahu, Syed Atif Ali

**Affiliations:** 1Department of Epidemiology & Public Health, Central University of Tamil Nadu, Thiruvarur 610005, India; sujitkumar@cutn.ac.in; 2Indian Veterinary Research Institute, Bareilly 243122, India; sabrinath.thankappan@icar.gov.in; 3Indian Council of Medical Research-National Institute of Epidemiology, Chennai 600077, India; 4Faculty of Veterinary and Animal Sciences, Rajiv Gandhi South Campus (RGSC), Banaras Hindu University (BHU), Mirzapur 231001, India; pkkm@bhu.ac.in; 5Institute of Biochemistry, Molecular Biology and Biotechnology, University of Colombo, Colombo 00700, Sri Lanka; rosh86niloo@gmail.com; 6Zoonosis Research Lab, Tamil Nadu Veterinary and Animal Sciences University, Chennai 600051, India; senthilkumar.k@tanuvas.ac.in; 7Livestock Economics and Statistics Division, ICAR-Indian Veterinary Research Institute (ICAR-IVRI), Bareilly 243122, India; mr.verma@icar.gov.in; 8Department of Animal Science, Centurion University of Technology and Management, Paralakhemundi 761211, India; abhishek.hota@cutm.ac.in; 9Biochemistry Laboratory, ICAR-Indian Veterinary Research Institute (ICAR-IVRI), Mukteswar 263138, India; s.sekar@icar.gov.in; 10School of Veterinary Medicine, Jimma University, Jimma 378, Ethiopia; yosef.deneke@ju.edu.et; 11Assistant Director General (ICAR), Krishi Bhawan, New Delhi 110001, India; ashok.kumar21@icar.gov.in; 12Indian Council of Medical Research-Regional Medical Research Centre, Port Blair 744103, India; na.muruganandam@gmail.com; 13National Institute of Research in Environmental Health, Bhopal 462030, India; DEEPANKER.DAS@rediffmail.com; 14Department of Microbiology, SCB Medical College, Cuttack 753007, India; drkhatua.sasmita@gmail.com; 15AVAS, Saraskana, Mayurbhanj 757051, India; radhakrishnasahu92@gmail.com; 16Institute of Chemistry, Academia Sinica, Taipei 115024, Taiwan; atifali_2413433@yahoo.com

**Keywords:** dot ELISA dipstick test, latex agglutination test, leptospirosis, microscopic agglutination test, point-of-care, sensitivity, specificity

## Abstract

Leptospirosis is a spirochaetal infection that possesses a broad host range affecting almost all mammals. In the present study, the microscopic agglutination test (MAT) was compared with recombinant LigA/B antigen-based point-of-care diagnostics such as the in-house IgM dot ELISA dipstick test (IgM- DEDT) and the latex agglutination test (LAT) for the serodiagnosis of human leptospirosis. The comparison of the MAT with these two point–of-care diagnostics was performed using the MAT as the gold standard test and using Bayesian latent class modelling (BLCM), which considers all diagnostic tests as imperfect. The N-terminal conserved region of the LigA/B protein spanning the first to fifth big tandem repeat domains (rLigA/BCon1-5) was employed as a serodiagnostic marker in both of the bedside assays. A total of 340 serum samples collected from humans involved in high risk occupations were screened using the MAT, IgM DEDT and LAT. During the early phase of leptospirosis, BLCM analysis showed that the IgM DEDT and LAT had similar sensitivities (99.6 (96.0–100)) and (99.5 (95.2–100)), respectively, while the single acute phase MAT had the lowest sensitivity (83.3 (72.8–91.3)). Both the IgM DEDT and the LAT may be superior to the single acute phase MAT in terms of sensitivity during the early phase of infection and may be suitable for the early diagnosis of leptospirosis. However, BLCM analysis revealed that the use of both acute and convalescent samples substantially increased the sensitivity of the final MAT (98.2% (93.0–99.8%)) as a test to diagnose human leptospirosis. Both the IgM DEDT and LAT can be employed as bedside spot tests in remote locations where the MAT is not easily accessible.

## 1. Introduction

Leptospirosis is a spirochaetal, zoonotic disease of ubiquitous distribution, with a higher incidence in tropical than in temperate regions [1]. Leptospirosis is presented with a wide spectrum of clinical manifestations such as pyrexia, jaundice, cephalgia, myalgia in the lower limbs, especially in the calves and thighs, conjunctival suffusion, epistaxis, haemoptysis, nausea, vomiting, constipation or diarrhoea, anorexia, abdominal pain and arthralgia [2,3]. Thus, leptospirosis, with its varied clinical manifestations, has been termed the ‘mysterious mimic’ by clinicians since it may mimic a large number of disease processes such as dengue, rickettsial infection, malaria and hantavirus infections, which makes laboratory diagnosis mandatory for this disease, especially in regions with a high incidence of other infections with a similar clinical picture [4,5].

A systematic review and modelling exercise estimated that there were annually 1.03 million cases of leptospirosis occurring worldwide with 58,900 reported deaths and leptospirosis was associated with the loss of 2.9 million disability-adjusted life years [6,7]. Human infection occurs due to exposure to environmental sources, such as animal urine, contaminated water or soil, or infected animal tissue [8]. In humans, the portals of entry include cuts or abraded skin, mucous membranes or conjunctivae [8]. Rarely, the infection may be acquired by the ingestion of food contaminated with urine or via aerosols [8]. In the tropics, endemic human leptospirosis is acquired through occupational exposure (subsistence farming) and living in rodent-infested, flood-prone, overcrowded urban areas [9]. Moreover, endemic leptospirosis is mainly a disease reported in humans with a low socio-economic status (including low education, poor housing, absence of sanitation and poor income) [10]. In humans, severe leptospirosis is most frequently, but not invariably, caused by serovars of the serogroup Icterohaemorrhagiae [11].

The microscopic agglutination test (MAT) is considered to be the gold standard serological test for detecting human leptospirosis [12]. However, the MAT has several inherent pitfalls such as the need for paired serum samples for detecting seroconversion, which is not practical in clinical settings as it delays disease diagnosis [13], a tedious and cumbersome test procedure involving live leptospiral antigens [14] and the need for verifying serovar identity regularly to ensure accurate results [15]. The MAT cannot distinguish between IgM antibodies indicative of a present infection and IgG antibodies indicative of a past infection [2]. Moreover, the MAT does not differentiate between infected and vaccinated animals (DIVA) since it cannot distinguish between antibodies formed due to a natural infection and those by vaccination [2]. Furthermore, the MAT can provide a false negative test result in the early course of the disease [2]. The MAT has also been reported to give false positive test results since cross-reactive antibodies in syphilis, relapsing fever, Lyme disease, enteric fever, dengue and malaria may give a titre of 1:80 or 1:100 [2].

The shortcomings of the MAT have forced disease investigators to search for alternative field-oriented tests [16]. The ideal diagnostic test for leptospirosis should be readily available, economical, should have high sensitivity and specificity during the acute phase of infection and give rapid test results [12]. The latex agglutination test (LAT) is a striking example of a highly economical, rapid screening, point-of-care test tailor-made for the large-scale screening of sera samples in endemic areas without using any sophisticated equipment [17]. The LAT has been used successfully in several instances by epidemiologists for the detection of anti-leptospiral antibodies in humans by coating latex beads with recombinant proteins such as LipL41, LipL32 and Lsa27 [17,18,19,20]. Another inexpensive and user-friendly point-of-care test widely used due to its simplicity is the Dot ELISA Dipstick test (DEDT) that can be used for detecting either IgM or IgG antibodies in humans and animals either in a single test format or for screening a large number of sera samples [21,22,23].

Recombinant outer membrane proteins (OMPs) such as LigA and LigB antigens, which are present exclusively in pathogenic and not saprophytic *Leptospira* species, have been reported to serve as serodiagnostic markers for acute phase leptospirosis [24]. The development of point-of-care diagnostic assays based on Lig proteins would permit early case detection and prevent disease progression and the severe outcomes associated with leptospirosis [23,24]. The N-terminal 630 amino acids of the LigA and LigB antigens (LigCon), covering the first six and a half Ig-like domains, are highly conserved between the two proteins [25]. Hence, in this study the first five conserved domains (sequence position K34 to S433) were chosen for expression in the heterologous *E. coli* system to serve as serodiagnostic markers for point-of-care tests such as the IgM DEDT and LAT.

Due to the inherent flaws of the MAT, several researchers have cast doubt on the validity of using it as an immunological gold standard for the comparative evaluation of rapid immunodiagnostics [12,26]. Bayesian latent class modelling, a statistical model that presumes that all diagnostic tests are imperfect, has been recommended as a more reliable method for evaluating rapid diagnostic tests, including immunodiagnostics for leptospirosis [26,27].

In this study, we compared two in-house point-of-care tests, such as the IgM DEDT detecting *LigBCon1-5 antigen*-specific IgM antibodies and the LAT detecting *LigBCon1-5 antigen*-specific IgM and IgG antibodies, with the MAT detecting both agglutinating IgM and IgG antibodies against a panel of sixteen leptospiral serovars. We analysed our findings using two statistical models, i.e., taking the MAT as the gold standard, and Bayesian latent class modelling, which considers each test as imperfect.

## 2. Material and Methods

### 2.1. Ethical Approval

The study was approved by the Institutional Human Ethics Committee (IHEC) of Indian Council of Medical Research-National Institute of Epidemiology (ICMR-NIE), Chennai (Ref. No. NIE/IHEC/201504-03). All the samples were collected from suspected patients showing clinical signs/symptoms. The delay between symptom onset and consultation among all the 340 patients was roughly 7 days. The tentative diagnosis for a suspected case of leptospirosis was defined based on the WHO-LERG epidemiological criteria [28], which include clinical signs and symptoms such as pyrexia, cephalgia, myalgia, especially in the lower limbs such as calf muscles and thighs, conjunctival suffusion, meningeal irritation, anuria, oliguria, proteinuria, jaundice, haemorrhages, arthralgia, skin rash, or a contact history of exposure to water or soil contaminated with urine of rodents or infected animals. Most of the samples were collected from humans involved in high risk occupations such as agriculture (rice and sugar cane cultivation) and animal husbandry activities (livestock farmers, butchers, veterinarians and rodent control workers). Blood samples were drawn after following aseptic procedures while keeping strict compliance with universal standard biosafety precautions. Blood samples were drawn from patients after obtaining written consent forms for voluntarily participating in the study.

### 2.2. Serum Sample Collection and Processing

A total of 340 human serum samples were collected from three Indian states viz. Odisha, Karnataka and Uttar Pradesh (U.P.). Follow-up (convalescent) samples were received from 179/340 patients. A volume of 5 mL of blood was drawn into serum collection tubes (Becton Dickinson, Franklin Lakes, NJ, USA) and the blood was allowed to clot for 1 h followed by centrifugation at 2000× *g* for 10 min. After centrifugation, the serum samples were stored at −20 °C for further laboratorial analysis. 

### 2.3. Leptospiral Serovars and Strains Used in Microscopic Agglutination Test (MAT)

A panel of 16 leptospiral serovars, namely, *Leptospira interrogans* serovar Australis strain Ballico, *L. interrogans* serovar Autumnalis strain Akiyami A, *L. interrogans* serovar Ballum strain S102, *L. interrogans* serovar Bataviae *strain* van *Tienen*, *L. interrogans* *serovar Canicola strain Hond Utrecht IV**, L. kirschneri* *serovar* Cynopteri *strain* 3522C, *L. interrogans* *serovar* Djasiman *strain* Djasiman, *L. kirschneri* *serovar* Grippotyphosa *strain* Moskva V, *L. borgpetersenii* *serovar* Hardjo *strain Hardjoprajitno*, *L. interrogans* *serovar* Hebdomadis *strain* Hebdomadis, *L. interrogans* *serovar* Icterohaemorrhagiae *strain* RGA, *L. borgpetersenii* *serovar* Javanica *strain* Veldrat Batavia 46, *L. noguchii* *serovar* Louisiana *strain* LSU 1945, *L. interrogans* *serovar* Pomona *strain* Pomona, *L. interrogans* *serovar* Pyrogenes *strain* Salinem and *L. borgpetersenii* *serovar* Tarassovi *strain* Perepelitsin were employed for performing MAT.

The leptospiral serovars used in MAT in this study were propagated by periodic sub-culture on a weekly basis of 0.5 mL of a one-week-old leptospiral culture into 5 mL of freshly prepared Ellinghausen–McCullough–Johnson–Harris (EMJH) media (Difco, Sparks, MD, USA) and incubated at 29 °C ± 1 °C in a stationary BOD incubator. The growth of Leptospira cultures were assessed by enumeration using dark field microscopy.

### 2.4. Microscopic Agglutination Test (MAT)

The agglutinins present in human sera against various serovars of *Leptospira* were screened using MAT following standard protocol [29]. The serum samples were diluted 1:50 in phosphate buffer saline (PBS), and various leptospiral antigens in volume equal to the diluted serum volume were added to each well, making the final serum dilution 1 in 100 in the screening test. In this study, 4–8-day-old live leptospiral antigens (approx. 2 × 10^8^ leptospires/mL) of 16 reference serovars were used. The micro-titre plates were incubated in stationary BOD incubator for 2 h and the serum–antigen mixtures were examined under a magnification of 20× using dark field microscope. The reciprocal of the highest serum dilution showing >50% reduction in the number of free non-agglutinable leptospires in the test when compared to the control for at least one leptospiral serovar was considered as the reporting titre. Single acute MAT seropositivity was defined as a titre of ≥400. Final/Total MAT seropositivity was defined as a titre of ≥400 in single sera sample, seroconversion from negative to a titre ≥ 100 or a four-fold rise in titre in paired (acute and convalescent) sera samples collected at a period of 3 weeks apart [3,28].

### 2.5. PCR Amplification and Cloning

Genomic DNA of *L. interrogans* serovar Pomona, encoding 1200 bp corresponding to first–fifth big tandem repeats of N-terminal conserved region of LigA/B gene without the signal sequence, was amplified by PCR using LigA/B F and LigA/B R primers (Eurofins Genomics India Pvt Ltd., Bengaluru, Karnataka, India). The primer sets, LigB F (5′-GGC GAG CTC AAA AGA GGC GGA GAT TCA TC-3′) and LigB R (5′-GGC AAG CTT AGA GTT ATC GTA AAA ATC CCG-3′), contained restriction sites for SacI and Hind III (underlined), respectively, in order to facilitate cloning into expression vector pQE30. PCR was performed for 30 cycles with initial denaturation at 95 °C for 4 min followed by denaturation at 94 °C for 1 min, annealing at 68 °C for 45 s, extension at 72 °C for 1 min and final extension at 72 °C for 5 min. Both the PCR amplified product and pQE30 vector (Qiagen, Hilden, North Rhine-Westphalia, Germany) were digested using SacI and Hind III restriction enzymes, and then PCR product was ligated to pQE30 vector using T_4_ DNA ligase. The newly constructed recombinant plasmid was designated as pQE30-LigBCon1-5 and was transformed into *Escherichia coli* (*E. coli*) M15 cells.

### 2.6. Induction of Expression and Purification of Recombinant LigA/BCon1-5 Antigen

The heterologous expression of recombinant LigA/BCon1-5 antigen was conducted in *Escherichia coli* (*E. coli*) M15 strain as per the method described previously [30]. Briefly, Luria Bertani broth (Difco) containing ampicillin at concentration of 100 µg/mL was used for growing *E. coli* M15 strain until the turbidity of the culture reached spectrometric reading of 0.5–0.7 at OD_600nm_. The heterologous expression in *E. coli* M15 strain was induced using 1 mM Isopropyl β-D-1-thiogalactopyranoside (IPTG) (Sigma-Aldrich, St. Louis, MO, USA) and the cells were allowed to grow further for 6 h at 37 °C. Sodium dodecyl sulfate–polyacrylamide gel electrophoresis (SDS-PAGE) was employed to analyse the yield of recombinant LigBCon1-5 fusion protein (rLigA/BCon1-5) by using standard protocol [31]. The denaturing agent 8 M Urea was used for extraction of the insoluble rLigA/BCon1-5 protein from inclusion bodies formed inside *E. coli* M15 strain. Purification of rLigA/BCon1-5 protein was performed using nickel–nitrilotriacetic acid (Ni-NTA) agarose affinity chromatography (Qiagen) as per manufacturer’s instructions.

### 2.7. Recombinant LigA/BCon1-5-Based IgM Dot ELISA Dipstick Test

IgM DEDT was performed according to the methods described previously [21,23] with minor modifications. IgM DEDT was standardized by dotting nitrocellulose membrane (NCM) (Thermo Fisher, Waltham, MA, USA) provided at the tips of plastic combs with 2 µL of rLigA/BCon1-5 antigen in carbonate–bicarbonate buffer in various amounts (250 ng to 1 µg) and allowed to be air dried for 30 min at room temperature. The unoccupied sites of the NCM were blocked with 3% bovine serum albumin in PBS-T (0.05% Tween-20 in PBS) overnight at 4 °C. These dotted NCM were washed with washing buffer comprising of 0.05% Tween-20 in PBS. Various dilutions of human sera, starting from 1:50 to 1:400, were diluted in blocking buffer and dispensed into wells of micro-titre plate (200 µL/well). The antigen-coated combs were dipped in the wells of the micro-titre plate and incubated for 1 h at 37 °C. Combs were then washed with washing buffer three times, each for duration of five minutes. After washing, the combs were dispensed in various dilutions (2500–10,000) of rabbit anti-human IgM HRP conjugate (Sigma-Aldrich) in blocking buffer and incubated for 1 h at 37 °C. The combs were then washed thrice with PBS-T and developed with 200 µL of substrate buffer containing 3,3′,5,5′-Tetramethylbenzidine (TMB) (Sigma-Aldrich) for 15 min in dark. The reaction was stopped by rinsing the strips with PBS. Appearances of blue colour dots on NCM indicate positive reaction. Non-appearance of blue colour dot on the nitrocellulose membrane indicate negative result. All the human sera samples were tested by including known positive and negative control.

### 2.8. Recombinant LigA/BCon1-5 Based Latex Agglutination Test

Latex beads (microspheres) were conjugated with rLigA/BCon1-5 antigen by passive adsorption as described previously [32] with minor modifications. A 2.5% suspension of dyed polystyrene microspheres of 1.0 µm diameter (Polysciences, Warrington, PA, USA) was washed thrice with glycine buffered saline (Glycine 0.1 M, NaCl 0.17 M; pH 8.2). Finally, the microspheres were made into a 2% suspension with glycine buffered saline, which was later mixed with an equal volume of rLigA/BCon1-5 antigen (1 mg/mL) diluted in the same buffer. The mixture was incubated overnight at room temperature in a shaking platform to ensure constant mixing. The sensitized latex beads were further blocked with 0.04% Bovine Serum Albumin (Difco, USA) and incubated overnight. Latex beads were centrifuged and the pellet was finally resuspended in glycine buffered saline as a 2% suspension containing 0.02% sodium azide. The conjugated latex beads were stored at 4 °C until use. LAT was performed on glass slides by mixing equal volume of serum (20 µL) and conjugated beads (20 µL). The result was read within 2 min. Samples were considered positive when there was formation of agglutination. Scores were designated as 3+, 2+ and 1+ to sera samples that agglutinated within 30 s, 30 s–1 min, 1–2 min, respectively, as described previously [33]. Samples were considered negative if no agglutination was observed.

### 2.9. Statistical Analysis

Statistical Package for the Social Sciences (SPSS) version 17.0 was used for performing statistical analysis. We assumed serum samples positive for MAT under two circumstances: (a) acute phase MAT (MAT during the acute/febrile phase of infection, a titre of ≥400), and (b) final MAT (either acute MAT, or a four-fold rise in MAT titre between acute and convalescent sera samples, or seroconversion on MAT to a titre of ≥100) as previously described [12]. For the purpose of gold standard analysis, human patients positive on ‘final MAT’ were considered true positives for leptospirosis. The sensitivity, specificity, positive and negative predictive values of IgM DEDT and LAT were calculated with the ‘final MAT’ as the gold standard. Finally, we compared both ‘acute MAT’ and ‘final MAT’ separately with IgM DEDT and LAT using Bayesian latent class modelling (BLCM). BLCM analysis was performed using Modelling for Infectious Disease Centre, Mahidol-Oxford Research Unit (MICE) tool [34].

## 3. Results

### 3.1. Microscopic Agglutination Test

The final MAT seropositivity of 19.41% (66/340) was reported for human leptospirosis with maximum agglutinins detected against serovar Icterohaemorrhagiae (*n* = 47) followed by Grippotyphosa (*n* = 26), Australis (*n* = 10), Hebdomadis (*n* = 03), Hardjoprajitno (*n* = 01) and Autumnalis (*n* = 01) as shown in Table 1. The cumulative figure of sera positive for various serovars (*n* = 88) exceeded the final MAT positive sera (*n* = 66) as several sera reacted with multiple leptospiral serovars. The MAT titre ranged from 1:100 to 1:1600. Antibodies to serovars Ballum, Bataviae, Canicola, Cynopteri, Djasiman, Javanica, Louisiana, Pomona, Pyrogenes and Tarassovi were absent in all the human sera against which the MAT was performed.

A total of 51 serum samples tested positive by the single acute MAT (MAT titre ≥ 400). Four human sera showed a four-fold rise in the MAT titre (≥ 50 to ≥ 200), while seroconversion (0 to ≥ 100) was shown by 11 human sera. The final MAT seropositivity levels for various Indian states such as Odisha, Uttar Pradesh and Karnataka were 21.76%, 18.0% and 15.71%, respectively (Figure 1a). The leptospiral serovar epidemiology found in these three Indian states are shown in Figure 1b–d.

### 3.2. Recombinant LigA/BCon1-5 Antigen Expression

A high-level expression of rLigA/BCon1-5 protein (expected and observed molecular weight on SDS-PAGE ~44 kDa and 51 kDa, respectively) was obtained after IPTG induction of *E. coli* M 15 cells harbouring recombinant plasmid pQE30-LigBCon1-5. The expression kinetics of rLigA/BCon1-5 antigen revealed that the protein first appeared on SDS-PAGE 2 h after IPTG induction and reached a maximum level at 8-h post induction (data not shown). The protein yield was approximately 20 mg/L following extraction of the insoluble rLigA/BCon1-5 protein from inclusion bodies using the denaturing agent 8 M Urea.

### 3.3. Recombinant LigA/BCon1-5-Based IgM Dot ELISA Dipstick Test

Standardization of the IgM-based DEDT was conducted by dotting the NCM provided at the tip of each plastic combs with 1 µg of purified rLigA/BCon1-5 antigen, and the optimum dilutions for field sera and anti-human IgM HRP conjugate were found to be 1:50 and 1:2500, respectively. The field sera, which gave blue colour dots on the NCM, were considered as positive while non-reactors failed to give a blue colour dot on the NCM (Figure 2). A positive correlation exists between the MAT titre and the intensity of colour of the blue dots. Human sera (*n* = 51), which showed a positive reaction in the single acute MAT (MAT titre ≥ 400), and human sera (*n* = 4), which showed a four-fold rise in the MAT titre (≥50 to ≥200), gave high intensity blue dots (Figure 2c–f), while human sera (*n* = 8), which showed seroconversion (0 to ≥100), gave low intensity blue dots (Figure 2a,b) on the NCM provided at the tips of plastic combs. Three human sera, which showed seroconversion (0 to ≥100), gave no blue colour dots (Appendix A), while one MAT^−ve^ sera gave a high intensity blue dot in the IgM DEDT (Appendix A).

### 3.4. Latex Agglutination Test and Correlation between MAT Titre and LAT Score

The latex agglutination test (LAT) showed a clear-cut agglutination with positive sera, which can be visualized easily (Figure 3b–d), and can clearly differentiate negative sera showing homogeneous suspension (Figure 3a). A positive correlation exists between the MAT titre and LAT score. Human sera (*n* = 51), which showed a positive reaction in the single acute MAT (MAT titre ≥ 400) gave a LAT score of 3+ (Appendix A), which meant that a high intensity of agglutination occurred almost instantaneously (usually <30 s) upon the addition of serum samples, which is evident in the form of tiny flakes (Figure 3d). Human sera (*n* = 4), which showed a four-fold rise in the MAT titre (≥50 to ≥200) gave a LAT score of 2+, and the agglutination occurred between 30 s and 1 min with a moderate intensity of agglutination in which the agglutinins formed a halo at the periphery during the process of swirling the glass slide, while the centre was virtually empty (Figure 3c). Human sera (*n* = 5), which showed seroconversion (0 to ≥100), gave a LAT score of 1+, and agglutination occurred in 2 min with a low intensity of agglutination (Figure 3b). Six human sera, which showed seroconversion (0 to ≥100), gave no agglutination in the LAT (Appendix A), while one MAT^−ve^ sera gave a 3+ LAT score (Appendix A).

### 3.5. Recombinant LigA/BCon1-5-Based IgM Dot ELISA Dipstick Test and Latex Agglutination Test Compared with MAT as Gold Standard

Using a single acute MAT as a gold standard, 15.0% (11.5–19.3%) of patients in the cohort had confirmed leptospirosis. Both the IgM DEDT and LAT had a sensitivity of 100% (91.3–100). However, the specificity of the LAT (96.5% (93.5–98.2%)) was slightly higher than that of the IgM DEDT (95.5% (92.2–97.5%)) (Table 2).

When a combination of acute and paired serum samples for the MAT (i.e., final MAT) was considered as a reference standard, the proportion of confirmed leptospirosis increased to 19.4% (15.4–24.1%). There was a significant reduction in the sensitivity of the LAT (90.9 (80.6–96.3%)). However, the IgM DEDT retained good levels of sensitivity (95.5% (86.4–98.8%)). However, both the IgM DEDT and LAT showed high specificity (99.6% (97.7–100.0%)) in comparison to the final MAT (Table 3).

### 3.6. Bayesian Latent Class Modelling for MAT, Recombinant LigA/BCon1-5-Based IgM Dot-ELISA Dipstick Test and Latex Agglutination Test

Using BLCM analysis, the proportion of patients diagnosed with leptospirosis among the group of patients using only acute sera samples (i.e., single acute MAT) was 0.18 (0.14–0.22). Using only acute sera samples, the sensitivities of the MAT, IgM DEDT and LAT were 83.3% (72.8–91.3%), 99.6% (96.0–100%) and 99.5% (95.2–100.0%), respectively, and specificities were 99.9% (99.1–100%), 98.9% (97.2–99.8%) and 99.9% (99.1–100%), respectively (Table 2).

Using BLCM analysis, the proportion of patients diagnosed with leptospirosis among the group of patients using both acute and paired sera samples (i.e., final MAT) was 0.19 (0.15–0.23). Sensitivities of the MAT, IgM DEDT and LAT in this group of patients were 98.2% (93.0–99.8%), 99.6% (96.0–100.0%) and 94.9% (87.8–98.6%), respectively; the specificities were 98.9% (97.2–99.7%), 99.9% (99.1–100.0%) and 99.9% (99.1–100.0%), respectively (Table 3).

## 4. Discussion

Leptospirosis is an important re-emerging disease worldwide that requires early and definitive diagnosis to guide the clinician to commence the appropriate treatment since antibiotic treatment is most effective when initiated early in the course of the disease [35,36]. Even though the MAT is widely considered as the immunological gold standard, our analysis using Bayesian latent class modelling (BLCM) showed that it has poor sensitivity (83.3% (72.8–91.3%)) when performed in the early phase of infection. However, the use of both acute and convalescent samples substantially increased the sensitivity of the MAT (98.2% (93.0–99.8%)) as a test to diagnose human leptospirosis. Our results are in concordance with the findings of a previous study conducted in Sri Lanka [12].

In low–middle income south-east Asian countries such as India, patients, especially low income daily wage labourers, who are forced to live from hand to mouth, rarely show up for convalescent sera collection once they have relief from symptoms following the administration of antibiotics. Hence, we devised the novel strategy of classifying the MAT results into two separate categories—single acute MAT and final MAT (acute/paired MAT). In this study, we separately compared the diagnostic accuracies of both the single acute MAT and the final MAT with two ‘point-of-care’ diagnostics for low-resource settings viz. the IgM DEDT (primary binding assay) and latex agglutination test (secondary binding assay).

Icterohaemorrhagiae was the predominant serovar observed in this study, which was the leading serovar in Odisha and Uttar Pradesh. The involvement of serovar Icterohaemorrhagiae indicates the contamination of waterlogged rice and sugar cane fields with rodent urine. Seroepidemiological studies in India conducted in rodents have pinpointed the involvement of serovar Icterohaemorrhagiae along with other serovars present in this study such as Australis, Autumnalis, Grippotyphosa and Hardjoprajitno to be associated with rodents such as *Rattus*, *Rattus norvegicus*, *Rattus hinton* and *Rattus rufescens* [37].

The LAT provides test results much faster (within two minutes) in comparison to the IgM DEDT, which has several steps in its procedure and takes about 4 h to perform. Our research team conducted a limited field trial using the rLigA/BCon1-5-based LAT in rural, resource-limited settings in India. The field trial revealed that LAT assay reagents possessed a long shelf life of at least 3–4 months in a tropical climate with high temperature and humidity, even with frequent electricity cuts in India affecting refrigeration temperatures, owing to the addition of 0.02% sodium azide as a preservative for sensitized latex bead suspension. Hence, this diagnostic assay is ideally suited for use at the peripheral level of the human health care system as a rapid screening test for leptospirosis.

The IgM DEDT and LAT are two ‘point-of-care’ diagnostics, which are tailor-made for low-resource settings since these tests meet the criteria of being user-friendly tests that can be easily performed, and test result interpretation can be performed by non-skilled personnel with minimal training. This is in stark contrast to the MAT, which is a cumbersome test due to the requirement of live leptospiral antigens whose maintenance is a tedious task that requires a dedicated and highly skilled technical staff with optimum training in handling live leptospiral cultures [16]. Moreover, both the IgM dot ELISA dipstick test and the LAT are highly economical bedside tests that require less sophisticated equipment [17,21], which is in sharp contrast to the MAT that requires costly equipment such as a biological oxygen demand (BOD) incubator and a dark field microscope (DFM) [23]. Additionally, both these spot tests generate limited amounts of biomedical waste and their alluring properties include their portability and simplicity [22,38], which is in clear contrast to the MAT that generates considerable biomedical waste, largely in the form of old leptospiral cultures that require a proper disposal system, and is non-portable due to the requirement of equipment and other laboratory facilities [23]. Hence, the IgM DEDT and LAT are serodiagnostic assays that are ideally suited for use at the tertiary level of the human health care system, especially in rural settings, which are mostly resource-constrained, as ‘point-of-care’ tests for the diagnosis of human leptospirosis.

The limitations of the two rapid diagnostic tests (RDTs), namely the IgM DEDT and LAT, are the poor sensitivity when compared with PCR (gold standard during the early course of infection) and the chance of obtaining false negative test results during the early course of the disease. Furthermore, if only these two RDTs are used without the MAT, the epidemiological data of infecting serovars and suspected environmental or animal reservoirs will not be available. One limitation of the present study is that the tertiary human health care centres where the two RDTs were tested had no PCR facilities since all of these health care centres were located in remote and rural locations of India.

## 5. Conclusions and Further Perspectives

Bayesian latent class modelling (BLCM) showed that the MAT has poor sensitivity, especially when performed in the early phase of infection. However, the use of both acute and convalescent samples substantially increased the sensitivity of the MAT. The novel strategy of classifying the MAT results into single acute MAT and final MAT (acute/paired MAT) helped to circumvent the problem associated with patients failing to provide convalescent sera once they obtain relief from symptoms following the administration of antibiotics. The recombinant LigA/BCon1-5 antigen proved to be an effective serodiagnostic marker when employed in primary (IgG dot ELISA dipstick test) and secondary binding assays (LAT) for the detection of human leptospirosis. In order to change the testing method practices to newer platforms, both the IgM DEDT and LAT should be employed as rapid, primary screening tests for detecting the early stages of leptospirosis and the MAT can be employed as a confirmatory test for recognising the epidemiological data of infecting serovars and suspected environmental or animal reservoirs in an endemic region. The use of both the IgM DEDT and LAT in rural settings would permit the implementation of intervention strategies based on early case detection and the timely initiation of treatment using antibiotics, which would prevent disease progression, thereby reducing the high mortality associated with leptospirosis.

## Figures and Tables

**Figure 1 diagnostics-12-01455-f001:**
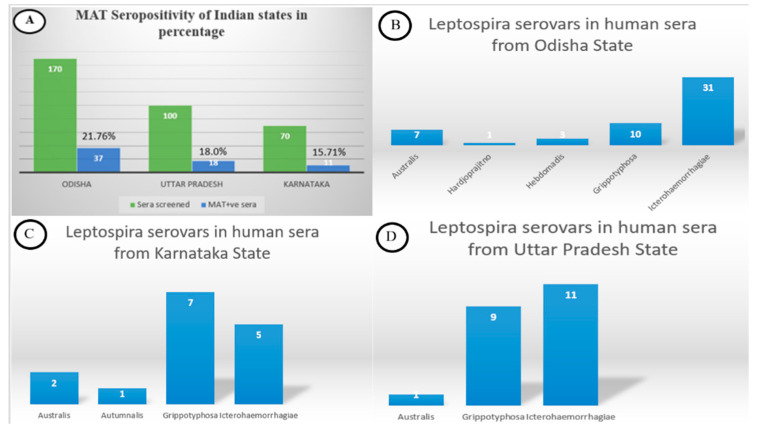
MAT seropositivity of Indian states in percentage and Leptospira serovar epidemiology in Indian states such as Odisha, Karnataka and Uttar Pradesh.

**Figure 2 diagnostics-12-01455-f002:**
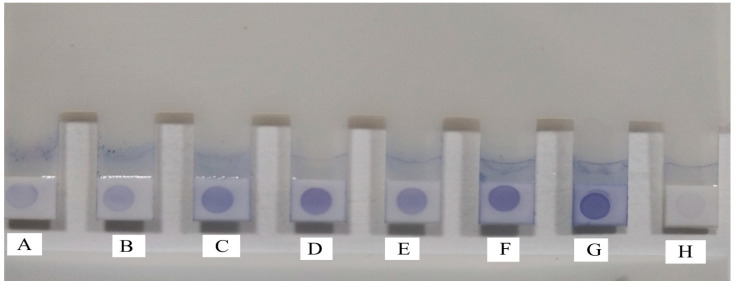
Recombinant LigA/BCon1-5 antigen-based IgM Dot ELISA Dipstick Test (IgM DEDT) for the detection of anti-leptospiral antibodies in human sera. (**A**,**B**): Low intensity blue dots by human sera which showed seroconversion (0 to ≥100); (**C**,**D**): Moderate intensity blue dots by human sera which showed a four-fold rise in the MAT titre (≥50 to ≥200); (**E**,**F**): High intensity blue dots by human sera which showed a positive reaction in the single acute MAT (MAT titre ≥ 400); (**G**): Known positive human sera (**H**): Known negative human sera.

**Figure 3 diagnostics-12-01455-f003:**
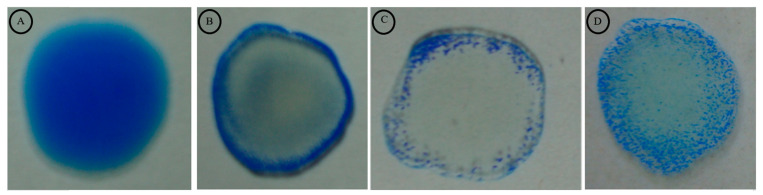
Recombinant LigA/BCon1-5 antigen-based latex agglutination test (LAT) for detection of anti-leptospiral antibodies in human sera. (**A**): Negative sera showing homogenous suspension (**B**): Human sera which showed seroconversion (0 to ≥100) giving a LAT score of 1+ve, (**C**): Human sera which showed a four-fold rise in the MAT titre (≥50 to ≥200) giving a LAT score of 2+ve, (**D**): Human sera which showed a positive reaction in the single acute MAT (MAT titre ≥ 400) giving a LAT score of 3+ve.

**Table 1 diagnostics-12-01455-t001:** Final MAT seropositivity at 1:100 cut off value showing agglutinins for various leptospiral serovars.

Genomo- Species	Serogroup	Serovar	Strain	Positive (N) *	Prevalence (%)	95% LCI	95% UCI
*L. interrogans*	Icterohaemorrhagiae	Icterohaemorrhagiae	RGA	47	13.82	10.55	17.89
	Australis	Australis	Ballico	10	2.94	1.60	5.33
	Hebdomadis	Hebdomadis	Hebdomadis	03	0.88	0.30	2.56
	Autumnalis	Autumnalis	Akiyami A	01	0.29	0.05	1.64
	Sejroe	Hardjo	Hardjoprajitno	01	0.29	0.05	1.64
*L. kirshneri*	Grippotyphosa	Grippotyphosa	Moskova V	26	7.65	5.27	10.97

* Cumulative figure of sera positive for various serovars (N = 88) exceed total MAT positive sera (*n* = 66) as several sera reacted with multiple leptospiral serovars.

**Table 2 diagnostics-12-01455-t002:** Comparison of the diagnostic accuracies of single acute MAT with IgM dot ELISA dipstick test and latex agglutination test by considering MAT as reference/gold standard as well as imperfect test as in BLCM.

Parameters	MAT Considered as Gold Standard Test (%)	Bayesian Latent Class Model (%)
**Prevalence**	15.0 (11.5–19.3)	18.0 (14.2–22.3)
**Acute MAT**		
Sensitivity	100	83.3 (72.8–91.3)
Specificity	100	99.9 (99.1–100)
PPV	100	99.5 (94.9–100)
NPV	100	96.5 (93.9–98.2)
**rLigA/BCon1-5-based IgM Dot ELISA Dipstick Test**		
Sensitivity	100 (91.3–100)	99.6 (96.0–100)
Specificity	95.5 (92.2–97.5)	98.9 (97.2–99.8)
PPV	79.7 (67.4–88.3)	95.2 (88.4–98.9)
NPV	100 (98.3–100)	99.9 (99.1–100)
**rLigA/BCon1-5-based LAT**		
Sensitivity	100 (91.3–100)	99.5 (95.2–100)
Specificity	96.5 (93.5–98.2)	99.9 (99.1–100)
PPV	83.6 (71.5–91.4)	99.6 (96.0–100)
NPV	100 (98.3–100)	99.9 (98.9–100)

**Table 3 diagnostics-12-01455-t003:** Comparison of the diagnostic accuracies of acute/paired MAT (final MAT) with IgM dot ELISA dipstick test and latex agglutination test by considering MAT as reference/gold standard as well as imperfect test as in BLCM.

Parameters	MAT Considered as Gold Standard Test (%)	Bayesian Latent Class Model (%)
**Prevalence**	19.4 (15.4–24.1)	18.9 (14.9–23.2)
**Acute MAT/Paired MAT**		
Sensitivity	100	98.2 (93.0–99.8)
Specificity	100	98.9 (97.2–99.7)
PPV	100	95.2 (88.4–98.8)
NPV	100	99.6 (98.3–100)
**rLigA/BCon1-5-based LAT**		
Sensitivity	90.9 (80.6–96.3)	94.9 (87.8–98.6)
Specificity	99.6 (97.7–100)	99.9 (99.1–100)
PPV	98.4 (90.0–99.9)	99.6 (95.9–100)
NPV	97.8 (95.1–99.1)	98.8 (97.1–99.7)
**rLigA/BCon1-5-based IgM Dot ELISA Dipstick Test**		
Sensitivity	95.5 (86.4–98.8)	99.6 (96.0–100)
Specificity	99.6 (97.7–100)	99.9 (99.1–100)
PPV	98.4 (90.5–99.9)	99.6 (96.2–100)
NPV	98.9 (96.6–99.7)	99.9 (99.1–100)

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
