# Peer review of "Diagnosis of Human Leptospirosis: Comparison of Microscopic Agglutination Test with Recombinant LigA/B Antigen-Based In-House IgM Dot ELISA Dipstick Test and Latex Agglutination Test Using Bayesian Latent Class Model and MAT as Gold Standard"

_diagnostics, 2022, doi:10.3390/diagnostics12061455_

Round 1

Reviewer 1 Report

Leptospirosis is a severe ane remerging disease. Its diagnosis is challenging for many reasons.

One of them is necessary specific tools to be used. Culture is fastidious, PCR is not available in many parts although it is now a gold standard for early diagnosis. MAT is designated as a gold standard but is difficult to perform especially in ressource limited settings.

First the authors should be commended for performing the design of new tools for such a neglected disease.

The authors designed and assessed 2 new rapid anti-leptospira antibody detection tests using a novel target namely a  recombinant LigA/BCon1-5 antigen . The authors used LAT and IgM DEDT methods. The conservation properties of the tests make them suitable for tropical use in remote settings.

The methods are clearly explained and written.

The results demonstrate a strong correlation between the MAT test observed and the results of the 2 RDTs and the intrinsec diagnostic performance seem high.

The main conclusion that can be drawn which is yet interesting is that these tests can substitute to MAT ( but without bringing serovar data). But they probably have the same limits than MAT. Indeed a major pitfall in leptospirosis diagnosis is the serological window ( grossly before the 6th day of fever or possibly later for MAT) during which most of the patients refer to health center. So the risk of false negative of the new RDTs still exist and has not been evaluated with patients having early samples and positive PCRs. These preliminary data are encouraging but does not demonstrate clinical value yet.

Finally these new tools could helps to have quickly a result but these data don’t support the help to have an early diagnosis. The authors should highlight these limits.

I have several comments to make to the authors

Methods

Could you be more precise on how were those 340 serum samples chosen ?

Could you bring more data regarding the patients especially how many patients were suspected of leptospirosis during the period (possibly without serological sample). Could you also bring data regarding the delay between symptoms onset and consultation.

L102 please give these results in appropriate results section

Results

Figure 1 is poorly readible, please remove the 3D effect and choose different colors to gain in clarity

Figure 3 the photos are blurred, could the authors provide better pictures ?

Table 2 : please rephrase the name of the column « MAT was assumed as a perfect gold standard (%)* »

Table 2 I suggest moving « rLigA/BCon1-5 based LAT » on the bottom of the table to correlate clearly with the order of the results presented in text

Table Could you precise what do the stars (*) refer to ?

Discussion

The findings on seropositivity rates according to the regions don’t add value here, please remove or at least shorten this part.

Please remove such detailed results in this section l 366-370

Even though PCR is not accessible in many parts of the world especially in rural areas could the authors discuss the fact that PCR has now turned in to a gold standard ? Did  they have any data regarding PCR that could bring additional information ?

The authors should point out more clearly the limits of the serological tests including these new RDTs during leptospirosis as they did for MAT in the intro section.. false negative during early course, poor sensitivity compared with PCR for example

The authors should add that if only RDTs only are used without any MAT,  the epidemiological data of infecting serovars and suspected environnemntal or animal reservoir will not be available anymore.

Reviewer 2 Report

Summary:

The manuscript titled “Diagnosing of Human Leptospirosis: Comparison of Microscopic Agglutination Test with Recombinant LigA/B Antigen based In-House IgM-Dot ELISA Dipstick Test and Latex Agglutination Test using Bayesian Latent Class Model and MAT as gold standard” seeks alternative immunoassays for the detection of human leptospirosis exposures. The study compares several serology platforms (IgM-DEDT, LAT) to the gold standard MAT using human serum samples collected in India. The manuscript provides strong data showing that the IgM-DEDT and LAT assays can be used at bedside tests to detect leptospirosis in rural areas of India. The figures and tables are clear and easy to read.  I have the following minor comments/suggestions:

TITLE:

Capitalize “gold standard”

ABSTRACT

Line 7: correct formatting in the word “that”. 

Lines 8-12: The sentence starting with “The aim…” is long and clunky. Consider splitting into two sentences.

INTRODUCTION

Line 45: Define DIVA

MATERIAL AND METHODS

Lines 111-121: Some of the species names are underlined and some are not. Is there a reason why particular species are underlined?

Line 143: Add a period at the end of the sentence.

RESULTS

Table 1: There is some highlighting in Genomo-species column. Please correct.

Figure 1-2. Are there a captions to the figures? Captions can be used to really emphasize the key take-away message of each figure. 

DISCUSSION

The authors did a great job summarizing the conclusions from their study. The data presented in this study supports the use of the two bedside assays as an alternative to the MAT.  Of interest in the discussion is the likelihood of using the IgM DEDT and LAT in rural areas. While the two assays are effective in detecting leptospirosis, do the authors anticipate the assays be widely used in rural areas where testing is already limited? Will these two assays have a significant impact on the diagnosis, and ultimately, the mitigation of leptospirosis infection in humans? Or will MAT continue to be used since it is considered the gold standard testing method? How does one change the testing method practices to newer platforms?

Reviewer 3 Report

Dear Authors, your manuscript titled "Diagnosis of Human Leptospirosis: Comparison of Microscopic Agglutination Test with Recombinant LigA/B Antigen based In-house IgM-Dot ELISA Dipstick Test and Latex Agglutination Test using Bayesian Latent Class Model and MAT as gold standard" is interesting because compares different test for the diagnosis of leptospirosis in humans. Unfortunately in the introduction, there is no data about the epidemiology of leptospirosis in humans and the major serotypes involved in the disease. I suggest you to implement these data in the text. In the conclusions please improve further perspectives.

Moreover, please emend some mistakes in the characters used (lines 7, 81, 82, al 2.3 paragraph. 
